# Isolation and Structure Elucidation of New Cytotoxic Macrolides Halosmysins B and C from the Fungus *Halosphaeriaceae* sp. Associated with a Marine Alga

**DOI:** 10.3390/md20040226

**Published:** 2022-03-25

**Authors:** Takeshi Yamada, Kanoko Yoshida, Takashi Kikuchi, Tomoya Hirano

**Affiliations:** 1Faculty of Pharmacy, Osaka Medical and Pharmaceutical University, 4-20-1, Nasahara, Takatsuki City 569-1094, Osaka, Japan; ompu52116533@s.ompu.ac.jp (K.Y.); tomoya.hirano@ompu.ac.jp (T.H.); 2Faculty of Pharmacy, Toho University, 2-2-1, Miyama, Funabashi City 274-8510, Chiba, Japan; takashi.kikuchi@phar.toho-u.ac.jp

**Keywords:** macrodiolide, halosmysins, *Halosphaeriaceae* sp., marine alga, cytotoxicity

## Abstract

Two new cytotoxic metabolites, halosmysins B and C, have been isolated from the fungus *Halosphaeriaceae* sp. OUPS-135D-4 separated from the marine alga *Sargassum thunbergii.* These chemical structures have been elucidated by 1D and 2D NMR, and HRFABMS spectral analyses. The new compounds had the same 14-membered macrodiolide skeleton as halosmysin A, which was isolated from this fungal strain previously. As the unique structural feature, a diketopiperazine derivative and a sugar are conjugated to the 14-membered ring of halosmysins B and C, respectively. The absolute stereostructures of them were elucidated by the chemical derivatization such as a hydrolysis, the comparison with the known compounds (6*R*,11*R*,12*R*,14*R*)-colletodiol and halosmysin A, and a HPLC analysis of sugar. In addition, their cytotoxicities were assessed using murine P388 leukemia, human HL-60 leukemia, and murine L1210 leukemia cell lines. Halosmysin B was shown to be potent against all of them, with IC_50_ values ranging from 8.2 ± 1.8 to 20.5 ± 3.6 μM, though these values were slightly higher than those of halosmysin A.

## 1. Introduction

Secondary metabolites produced from marine-derived microorganisms have diverse structures and exhibit unexpected biological activities [1,2,3,4]. This study evaluates marine-derived fungi as a seed source for antitumor chemotherapeutic agents. Our group has published papers on the exploratory research of fungal metabolites [5]. Recently, we isolated a cytotoxic compound, halosmysin A (**1**), with a unique skeleton with a thiosilvatin analog conjugated to a 14-membered macrodiolide from the *Halosphaeriaceae* sp. OMPU-135D-4 strain separated from the marine alga *Sargassum thunbergii.* (Figure 1) [6]. The related polyketide macrodiolides such as colletodiol (**4**)**,** colletoketol (grahamimysins A), colletoll, and colletallol from *Colletotrichum capsici* [7]; grahamimysins A_1_ and B from *Cytospora* sp. [8,9,10]; 9,10-dihydrocolletodiol from *Varicosporina ramulosa* [11]; clonostachydiol from *Clonostachys cylindrospora* [12], *Gliocladium* sp. [13], and *Xylaria* sp. [14,15]; 4-keto-clonostachydiol from *Gliocladium* sp. [13,16]; cordybislactone from *Cordyceps* sp. [14]; acremonol and acremodiol from anunidentified *Acremonium*–like anamorphic fungus [17]; and acaulones from *Acaulium* sp. [18] exhibited diverse activities such as antimicrobial, antiosteoporosis, tyrosine kinase inhibition, anthelmintic, and cytotoxicity. The stereochemistry of these macrodiolides, consisting of two unsymmetrical subunits, was grouped into two types: the colletodiol-type, possessing 6*R*,14*R* and the clonostachydiol-type, possessing 6*S*,14*S* (Appendix A). In a previous study, the absolute configuration of **1** was established to be 6*R*,14*R* upon a comparison with known compound **4** [6]. In the present study, an ongoing search for cytotoxic metabolites from this strain led to the isolation of two new 14-membered macrodiolides designated as halosmysins B (**2**) and C (**3**) (Figure 1). The structural analysis of **2** by ^1^H and ^13^C nuclear magnetic resonance (NMR) spectroscopic techniques showed that **2** has the same planar structure as **1**. On the other hand, the alkaline hydrolysis of **3** gave the enantiomer of 5-hydroxy-(2*E*)-hexenoic acid derived from **1** and **4**. This paper reports the absolute stereostructures of **2** and **3**, including nuclear Overhauser effect spectroscopy (NOESY) data, information garnered from the *J* values in the ^1^H NMR spectrum, and a plausible biosynthetic proposal. The cytotoxic activities of these compounds against the murine P388 leukemia, human HL-60 leukemia, and murine L1210 leukemia cell lines are also described.

## 2. Results and Discussion

*Halosphaeriaceae* sp., a fungal strain from *S. thunbergii*, was incubated at 27 °C for six weeks in a medium (70 L) containing 1% glucose, 1% malt extract, and 0.05% peptone in artificial seawater adjusted to pH 7.5. The EtOAc extract of the culture filtrate after incubation was purified on a silica gel column, followed by reverse-phase high-performance liquid chromatography (HPLC), affording halosmysins B (**2**) (0.8 mg) and C (**3**) (5.5 mg). The structural determination of halosmysin A (**1**) has been reported [6], and (6*R*,11*R*,12*R*,14*R*)-colletodiol (**4**) was identified by comparison with the data reported by MacMillan et al. [7].

Halosmysin B (**2**) had the formula C_31_H_38_N_2_O_9_S established by *m*/*z* 615.2378 [M + H]^+^ (calcd for C_31_H_39_N_2_O_9_S: 615.2376) by high-resolution fast atom bombardment mass spectrometry (HRFABMS) (Appendix A). The Fourier transform infrared spectrum revealed peaks at 3378 and 1712 cm^−1^, which were assigned to the stretching vibrations of hydroxy and carbonyl groups, respectively (Appendix A). An inspection of the ^1^H and ^13^C NMR spectra (Table 1 and Appendix A, Appendix A) of **2** using distortionless enhancement by polarization transfer (DEPT) and ^1^H-^13^C correlation spectroscopy (HSQC) showed the presence of the following: one thiomethyl group (3′-SCH_3_); two secondary methyls (C-15 and C-16); two olefinic methyls (C-15′ and C-16′); four sp^3^-hybridized methylenes (C-5, C-13, C-7′, and C-12′); one of which is an oxygen-bearing sp^3^-methylene (C-12′); five sp^3^-methines (C-6, C-9, C-10, C-12, and C-14); three of which are oxygen-bearing sp^3^-methines (C-6, C-12, and C-14); seven sp^2^-methines (C-3, C-4, C-9′, C-10′, and C-13′); two quaternary sp^3^-carbons (C-3′, and C-6′); three quaternary sp^2^-carbons (C-8′, C-11′ and C-14′); and five carbonyl groups (C-2, C-8, C-11, C-2′, and C-5′). The correlations observed in ^1^H-^1^H correlation spectroscopy (COSY) and the ^1^H-^13^C heteronuclear multiple bond correlation (HMBC) spectra showed that **2** had the same planar structure as **1**, which was a 14-membered macrodiolide conjugated to a thiosilvatin analog, that is, a 3,6-bis(methylthio)-2,5-piperazinedione derivative (Appendix A). The HMBC correlations from H-1′ (NH) to C-3′and C-7′, from H-4′ (NH) to C-6′, from S-CH_3_ to C-3′, from H-7′ to C-5′, C-6′, C-8′, and C-9′, from H-9 to C-6′, from H-10 to C-3′, and H-7′ to C-9 suggested that the 14-membered bislactone moiety and the diketopiperazine derivative were bound between C-9 and C-6′ and between C-10 and C-3′ (Appendix A). The NMR chemical shifts of these signals closely resembled those of **1** except for the NMR signals for the thiosilvatin moiety [proton signals—H-9 (*δ*_H_ 3.59 d), H-10 (*δ*_H_ 4.81 d), H-7′ (*δ*_H_ 2.99 d, 3.56 d), and H-10′ (*δ*_H_ 7.30 d); carbon signals—C-9 (*δ*_C_ 52.6), C-10 (*δ*_C_ 52.4), C-5′ (*δ*_C_ 169.5), and C-6′ (*δ*_C_ 62.2)]. This suggested that **2** was a stereoisomer of **1** around the thiosilvatin moiety.

The relative configuration and the conformation of **2** were investigated by NOESY experiments (Appendix A and Figure 2 and Appendix A). For the 14-membered ring moiety of **2**, the NOESY correlations between H-3 and H-5β and between H-4 and H-6, and the large coupling constants (*J*_5β,6_ = 12.6 Hz and *J*_5β,4_ = 10.8 Hz) showed that the angle between bond C-6–C-5 and bond C-4–C-3 was as that of **1**. Furthermore, the NOESY correlations from H-16 to H-13α and H-13β and the large coupling constants (*J*_13β,14_ = 11.4 Hz) revealed the angle between bond C-14–C-16 and bond C-13–C-12 (Figure 2 and Appendix A). The NOESY correlations between H-9 and H-4′ (NH) and between H-10 and H-4′ (NH) were observed in **1**. However, the correlation observed in **2** was only between H-10 and H-4′ (NH), and H-9 correlated with H-1′ (NH). The NOESY correlations (H-9/H-3, H-10/H-12, and H-10/3′-SCH_3_) suggested that the orientation for H-10 was opposite to that of H-9. The above evidence showed that the stereo-arrangements of H-3′ and H-6′ in the diketopiperazine ring of **2** were opposite to those of **1**. Thus, the relative configuration of **2** was established, as shown in Figure 2, which was a stereoisomer of **1** at C-10, C-3′, and C-6′. As described elsewhere [6], the absolute configuration of **1** was determined by alkaline hydrolysis, i.e., the ^1^H NMR spectrum and the specific rotation of the reaction product, 5-hydroxy-(2*E*)-hexenoic acid, were identified with those of the hydrolyzed product from (6*R*,11*R*,12*R*,14*R*)-colletodiol (**4**). The reaction gave (–)-5-hydroxy-(2*E*)-hexenoic acid when the same procedure was applied to **2**, as expected. Hence, the absolute stereostructure of **2** was 6*R*,9*S*,10*R*,12*R*,14*R*,3′*S*,6′*R* (Figure 2).

In a previous study, we hypothesized the biosynthetic pathway of **1** from colletoketol, which was derived from **4** via the oxidative process [6]. The 3*R*-derivative shown in Figure 1, which could be formed by the elimination of the SCH_3_ group in a 3*R*,6*R*-bis(methylthio)-2,5-piperazinedione derivative, such as *cis*-bis(methylthio) silvatin [19], could be attacked by the *π*-electron at C-9 in colletoketol, forming a bond between C-9 and C-6′ in **1**. The C-10 in the macrodiolide could be attacked from below by C-3 in 3*R*-piperazinedione, forming a bond between C-10 and C-3′ in **1** (Figure 1). On the other hand, **2** could be formed by the attack of C-3 in the 3*S*-derivative formed from a 3*S*,6*S* or 3*S*,6*R*-bis(methylthio)-2,5-piperazinedione derivative, such as Sch 54,794 [20,21] or *trans*-bis(methylthio)silvatin [20], to above of C-10 in colletoketol. This final nucleophilic attack was on the opposite side of that forming **1**.

Halosmysin C (**3**) was assigned the molecular formula C_20_H_30_O_11_ from HRFABMS (Appendix A). The ^1^H and ^13^C NMR spectra of **3** (Table 1 and Appendix A, and Appendix A) showed the signals for the 14-membered macrodiolide skeleton, which were similar to those of **1** and **2**, except for that the signals for the double bond [proton signals; H-9 (*δ*_H_ 6.08 dd) and H-10 (*δ*_H_ 6.77 dd), carbon signals; C-9 (*δ*_C_ 125.2) and C-10 (*δ*_C_ 147.2)] were observed. Therefore, **3** was derived from colletodiol (**4**). For the partial molecule conjugated to the macrodiolide skeleton, the signals for the bis(methylthio)silvatin derivative observed in **1** and **2** disappeared. However, those for a sugar [the proton signals—H-1′ (*δ*_H_ 5.00 d), H-2′ (*δ*_H_ 3.43 dd), H-3′ (*δ*_H_ 3.67 dd), H-4′ (*δ*_H_ 3.34 d), H-5′ (*δ*_H_ 3.55 dt), and H-6′ (*δ*_H_ 3.64 d); the carbon signals—C-1′ (*δ*_C_ 102.9), C-2′ (*δ*_C_ 73.9), C-3′ (*δ*_C_ 75.1), C-4′ (*δ*_C_ 71.4), C-5′ (*δ*_C_ 74.3), and C-6′ (*δ*_C_ 62.1)] appeared. The HMBC correlation from H-11 to the anomeric carbon C-1′ showed the sugar was conjugated to C-11 in **3**. The correlations (H-2′/H-4′ and H-3′/H-5′) in the NOESY experiment and the coupling constant (*J*_1′,2′_ 3.6, *J*_2′,3′_ 9.6, *J*_3′,4′_ 9.6, and *J*_4′,5′_ 10.2 Hz) in the ^1^H NMR spectrum in **3** showed that the sugar in this molecular structure was α-glucose (Figure 3 and Appendix A).

Alkaline hydrolysis, which was the same procedure used with **1** and **2**, was carried out to determine the absolute configuration of **3**. The purification of the reaction mixture gave two carboxylic acids. The ^1^H NMR spectral data of them were in perfect agreement with those of the carboxylic acids obtained by the hydrolysis of **4**, 5-hydroxy-(2*E*)-hexenoic acid and 4,5,7-trihydroxy-(*2E*)-octenoic acid, respectively (Figure 2). Furthermore, the specific rotations of 4,5,7-trihydroxy octenoic acid obtained from **3** and **4** were in agreement. However, the specific rotations of **3**-derived 5-hydroxy hexenoic acid showed a positive sign ([α]_D_ + 9.1), and that of **4**-derived showed a negative sign ([α]_D_ − 14.3). The evidence showed that the absolute configuration in the 14-membered macrodiolide moiety of **3** is 6*S*,11*R*,12*R*,14*R*. The absolute stereostructure of the sugar conjugated to C-11 in the 14-membered ring was determined by the discrimination between the aldose enantiomers using reversed-phase HPLC. In producing the standard samples, D- and L-glucose were treated with L-cysteine methyl ester hydrochloride and *o*-torylisothiocyanate in pyridine, respectively. Each reaction mixture was analyzed by HPLC, which showed retention times for the D-glucose and L-glucose derivatives of 17.7 min and 12 min, respectively [22]. After the hydrolysis by hydrochloric acid of **3**, the same procedure as standard gave the HPLC peak at 17.7 min. In addition, the ^1^H NMR spectral data of the sugar derivative from **3** were in good agreement with those of the D-glucose derivative. Therefore, **3** was confirmed to be the glycoside with α-D-glucose.

As a primary screen for the antitumor activity, the cancer cell growth inhibitory properties of halosmysins B (**2**) and C (**3**) isolated in this study were examined using murine P388 leukemia, human HL-60 leukemia, and murine L1210 leukemia cell lines. A previous study reported that **1** had potent cytotoxicity against these cell lines, whereas **4** did not inhibit cell growth [6]. As expected, **2**, having the same piperazinedione derivative, showed potent activity comparable to 5-fluorouracil against all these cells, particularly the HL-60 cell line. Compound **3**, colletodiol glycoside, did not inhibit cell growth (Table 2). The addition of various functional groups to the double bond between C-9 and C-10 provides information on the structure-activity relationship and the detailed mechanism of activity. Therefore, the search for 14-membered macrolide analogs from this fungal metabolite will continue.

## 3. Materials and Methods

### 3.1. General Experimental Procedures

These are the same procedures as those in recent reports [6]. NMR spectra were recorded on an Agilent-NMR-vnmrs (Agilent Technologies, Santa Clara, CA, USA) 600 and 400 with tetramethylsilane (TMS) as an internal reference. FABMS was recorded using a JEOL JMS-7000 mass spectrometer (JEOL, Tokyo, Japan). IR spectra were recorded on an IRAffinity-1S (Shimadzu, Kyoto, Japan). Optical rotations were measured using a JASCO DIP-1000 digital polarimeter (Tokyo, Japan). Silica gel 60 (230–400 mesh, Nacalai Tesque, Inc., Kyoto, Japan) was used for column chromatography with medium pressure. ODS HPLC was run on a JASCO PU-1586 (Tokyo, Japan) equipped with a differential refractometer RI-1531 (Tokyo, Japan) and Cosmosil Packed Column 5C18-MSII (25 cm × 20 mm i.d., Nacalai Tesque, Inc., Kyoto, Japan). Analytical TLC was performed on precoated Merck aluminum sheets (DC-Alufolien Kieselgel 60 F254, 0.2 mm, Merck, Darmstadt, Germany) with the solvent system CH_2_Cl_2_–MeOH (19:1) (Nacalai Tesque, Inc., Kyoto, Japan), and compounds were viewed under a UV lamp (AS ONE Co., Ltd., Osaka, Japan) and sprayed with 10% H_2_SO_4_ (Nacalai Tesque, Inc., Kyoto, Japan) followed by heating.

### 3.2. Fungal Material

The fungus *Halosphaeriaceae* sp. was isolated from the surface of the marine alga *Sargassum thunbergii. Halosphaeriaceae* sp. collected at Osaka bay, Japan in July 2017. The fungal strain was identified based on the result of ITS rDNA nucleotide sequence analysis by Techno Suruga Laboratory Co., Ltd. (Shizuoka, Japan). The alga *Sargassum thunbergii* was wiped with EtOH and a cutting was applied to the surface of nutrient agar layered in a Petri dish. Serial transfers of one of the resulting colonies provided a pure strain of *Halosphaeriaceae* sp.

### 3.3. Culturing and Isolation of Metabolites

The fungus was cultured at 27 °C for four weeks in a medium (70 L) containing 1% glucose, 1% malt extract, and 0.05% peptone in artificial seawater adjusted to pH 7.5. Then, the culture filtrate was extracted thrice with AcOEt. The combined extracts were evaporated *in vacuo* to afford a mixture of crude metabolites (12.2 g). The EtOAc extract was chromatographed on a silica gel column with a CH_2_Cl_2_/MeOH gradient as the eluent to afford Fr. 1 (1% MeOH in CH_2_Cl_2_ eluate,188.5 mg) and Fr. 2 (5% MeOH in CH_2_Cl_2_ eluate, 24.7 mg). Fr. 1 was purified by ODS HPLC using MeOH/H_2_O (90:10) as the eluent to afford **4** (9.7 mg) and Fr. 3 (8.8 mg). Fr. 3 was purified by ODS HPLC using MeOH/H_2_O (80:20) as the eluent to afford **2** (0.8 mg) and **1** (5.7 mg). Fr. 2 was purified by ODS HPLC using MeOH/H_2_O (60:40) as the eluent to afford **3** (5.5 mg). These compounds were supplemented by several cultures to use in the following reactions and assays.

Halosmysin B (**2**): pale yellow oil; ${\left[\alpha \right]}_{D}^{22}$ + 5.6 (*c* 0.06, CHCl_3_); IR (neat) *ν*_max_/cm^−1^: 3378, 2925, 1712, 1665, 1601, 1512. HRFABMS *m*/*z* 615.2378 [M + H]^+^ (calcd for C_31_H_39_N_2_O_9_S: 615.2376); NMR data, see Table 1 and Appendix A.

Halosmysin C (**3**): pale yellow oil; ${\left[\alpha \right]}_{D}^{22}$ + 16.3 (*c* 0.041, EtOH); IR (neat) *ν*_max_/cm^−1^: 3420, 2925, 1706, 1652. HRFABMS *m*/*z* 469.1692 [M + H]^+^ (calcd for C_20_H_30_O_11_Na: 469.1680); NMR data, see Table 1 and Appendix A.

### 3.4. Alkaline Hydrolysis of **2**

Using the same procedure of **1** [6], **2** (1.2 mg) and 0.1 N NaOH aq. (1 mL) were stirred at room temperature for 14 h, then acidified with conc. HCl and extracted with AcOEt. The organic layer was evaporated in vacuo, and the residue was purified by ODS HPLC using MeOH/H_2_O (0.1% AcOH) (30:70) as the eluent to (–)-5-hydroxy-(2E)-hexenoic acid (0.4 mg) (r.t. 27.5 min).

(–)-5-hydroxy-(2E)-hexenoic acid: clear oil; ${\left[\alpha \right]}_{D}^{22}$ − 13.9 (c 0.06, EtOH); ^1^H NMR (400 MHz, MeOH-d_4_) δ ppm: 1.18 (3H, d), 2.31 (2H, dd), 3.85 (1H, sext), 5.82 (1H, d), 6.95 (1H, dt).

### 3.5. Alkaline Hydrolysis of **3**

Using the same procedure of **1** [6], **3** (2.2 mg) and 0.1 N NaOH aq. (1 mL) were stirred at room temperature for 4 h, then acidified with conc. HCl and extracted with AcOEt. The organic layer was evaporated in vacuo, and the residue was purified by ODS HPLC using MeOH/H_2_O (0.1% AcOH) (30:70) as the eluent to (+)-4,5,7-trihydroxy-(2E)-octenoic acid (0.9 mg) (r.t. 14.5 min) and (+)-5-hydroxy-(2E)-hexenoic acid (0.3 mg) (r.t. 27.2 min).

(+)-4,5,7-trihydroxy-(2E)-octenoic acid; clear oil; ${\left[\alpha \right]}_{D}^{22}$ 11.0 (c 0.04, EtOH); ^1^H NMR (400 MHz, MeOH-d_4_) δ ppm: 1.20 (3H, d), 1.60 (2H, dd), 3.75 (1H, m), 3.96 (1H, dt), 4.18 (1H, dd), 6.05 (1H, d), 7.02 (1H, dd).

(+)-5-hydroxy-(2E)-hexenoic acid; clear oil; ${\left[\alpha \right]}_{D}^{22}$ 9.1 (c 0.11, EtOH).

### 3.6. Sugar Analysis of **3**

Glycoside **3** was analyzed with reference to the method described in Ref. [22]. **3** (4.8 mg) and 0.5 N HCl (1 mL) were stirred at 80 °C for 2 h. The reaction mixture was evaporated in vacuo, and the residue was dissolved in pyridine (1 mL) containing L-cysteine methyl ester hydrochloride (5.0 mg), and was heated at 60 °C for 1 h. *o*-torylisothiocyanate (5.0 mg) was added to the reaction mixture, and was heated at 60 °C for 1 h. The reaction mixture was directly analyzed by ODS HPLC using MeOH/H_2_O (70:30). The peak at 17.7 min was coincident with derivative of D-glucose.

### 3.7. Assay for Cytotoxicity

Cytotoxic activities of **1**, **2** and **3** were examined by the same procedure to date [6], the 3-(4,5-dimethyl-2-thiazolyl)-2,5-diphenyl-2H-tetrazolium bromide (MTT) method. P388, HL-60, and L1210 cells were cultured in RPMI 1640 Medium (10% fetal calf serum) at 37 °C in 5% CO_2_. The test materials were dissolved in dimethyl sulfoxide (DMSO) to give a concentration of 10 mM, and the solution was diluted with the Essential Medium to yield concentrations of 200, 20, and 2 μM, respectively. Each solution was combined with each cell suspension (1 × 10^5^ cells/mL) in the medium, respectively. After incubating at 37 °C for 72 h in 5% CO_2_, grown cells were labeled with 5 mg/mL MTT in phosphate-buffered saline (PBS), and the absorbance of formazan dissolved in 20% sodium dodecyl sulfate (SDS) in 0.1 N HCl was measured at 540 nm with a microplate reader (MTP-310, CORONA electric). Each absorbance value were expressed as a percentage relative to that of the control cell suspension that was prepared without the test substance using the same procedure as that described above. All assays were performed three times, and semilogarithmic plots were constructed from the averaged data, and the effective dose of the substance required to inhibit cell growth by 50% (IC_50_) was determined.

### 3.8. The Origin of the Cell Lines

The P388 cell line was obtained from Dr. Numata (death, Osaka Medical and Pharmaceutical University, Japan). The HL-60 cell line was obtained from Dr. Kawai (death, Fuso Pharmaceutical Industries, Ltd., Osaka, Japan). The L1210 cell line was from Dr. Endo (Kanazawa University, Japan).

## 4. Conclusions

In conclusion, we have isolated two new cytotoxic metabolites, halosmysins B and C, from the marine-alga-derived fungus *Halosphaeriaceae* sp., which had the same 14-membered macrodiolide skeleton as halosmysin A. We determined the absolute configuration of them using the chemical technique. **2** exhibited a potent cytotoxicity against the HL-60 cell line. In order to study the structure-activity relationship and the biosynthetic pathway, the search for 14-membered ring macrolide analogs from this fungal metabolite will be continued.

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
