# Peer review of "Isolation and Structure Elucidation of New Cytotoxic Macrolides Halosmysins B and C from the Fungus Halosphaeriaceae sp. Associated with a Marine Alga"

_marinedrugs, 2022, doi:10.3390/md20040226_

Round 1

Reviewer 1 Report

1. Since the manuscript is a segmented publication in relation to the first paper on Halosmycin A, I suggest to improve the title of the manuscript. The title is almost identical to the first paper  "Halosmysin A, a Novel 14-Membered Macrodiolide Isolated from the Marine-Algae-Derived Fungus Halosphaeriaceae sp." except that in this paper it is "Halosmysins B and C" with the word "new cytotoxic" added in the title.  Perhaps the authors revise the title to make it different from the previous paper, e.g., focus on the novelty of its putative biosynthetic pathway and others.

2. Abstract line 20 indicates that halosmycin B is "very potent against all cell line tested" but looking at the data in Table 2 line 184, it is Halosmycin A that has stronger cytotoxic activity among the three analogues. Please correct this statement. 

2. The structure elucidation methodology, presentation of data, and data analysis were all thoroughly described.  However, I would like to see  in the supporting information the 

a) HPLC chromatogram showing the three related Halosmycins (A, B, and C) eluted in different elution times

b) 1H NMR of (-)-5-hydroxy-(2E)-hexenoic acid

c) 1H NMR  (+)-4,5,7-trihydroxy-(2E)-octanoic acid

3. Is the ITS rDNA nucleotide sequence of fungus Halosphaeriaceae deposited in a data base? If yes, kindly include the accession number in the methodology, section 3.2.

Author Response

  1. The title has been changed with reference to reviewer’s suggestion as below.

“Structural determination of new macrolides, halosmysins B and C, isolated from the marine-alga-derived fungus, and the consideration of the biosynthetic pathway of halosmysins A and B”

  1. For the description of the activity of halosmysins B (page1, line 20), we have added the following sentence to the end: “though these values were slightly higher than those of halosmysin A”.
  2. a) HPLC chromatogram, b) PMR of (-)-5-hydroxy-(2E)-hexenoic acid, and c) PMR of (+)-4,5,7-trihydroxy-(2E)-octanoic acid were added to the supporting information
  3. There is no deposition information from the classification appraisal request destination. Therefore, the accession number does not exist.

Reviewer 2 Report

The manuscript reports on the isolation and structural characterization of novel cytotoxic macrolides isolated from cultures of a fungal strain derived from a marine alga and identified as Halosphaeriaceae sp.

The structural characterization of the new compounds, including absolute configuration, relies on the analysis of MS, IR and NMR spectral data (1D and 2D), in comparison with that of structurally related compounds also isolated and characterized from the same authors, and on some chemical degradation and derivatization.

The analysis of NMR spectral data is accurate and clearly described and appears to be consistent with the proposed structures, with stereochemistry supported by chemical degradation, derivatization, and comparison with reference compounds.

The experimental procedures are well described and appear to be appropriate to the scope of the manuscript.

The cytotoxic activity against selected tumor cell lines at micro-M level, disclosed for some compound by biological assays, although limited is of some relevance and might support further investigation.

Extensive set of experimental data is presented in supporting materials.

A few suggestions are listed below:

1) insert the names of the compounds in Figure 1

2) page 6 lines 151 and 152 and page 8 line 241: check spelling of “4,5,7-thrihydroxy-(2E)-octenoic acid”

3) page 8 lines 226 and 235: correct “preceudure” in “procedure”

4) page 8 lines 250 and 252: correct “o-torylisothiocyanate” and “was coincident”

Some English editing might be required

Author Response

1) The names of the compounds were inserted to the title of Figure 1.

2) Thank you for your pointing out, we corrected “octanoic” to “octenoic”.

3) Thank you for your pointing out, we corrected “preceudure” to “procedure”.

4) Thank you for your pointing out, we corrected “torylisothiocyanat” to “torylisothiocyanate”, and “was coincided” to “was coincident”.

Reviewer 3 Report

This manuscript described the isolation and structural elucidation of two new  macrodiolides and two known analogues with a 14-membered ring from a strain of marine-alga derived fungus Halosphaeriaceae sp. The compounds were evaluated for their cytotoxicities against three tumor cell lines.

However, the two new compounds are similar to the previously reported analogues halosmysin A and colletallol, and their cytotoxocities are not potent (only weak to moderate), indicating a low novelty and significance of this work.

The authors tried to describe the structure-activity relationship (SAR) for these macrolides, but the SAR can not be derived based on a limited number of compounds. Thus, the conclusion is not convincing enough.

Bsed on the reasons above, this manuscript is unpublishable in Marine Drugs

Author Response

As be pointed out, the structural novelty of new compounds in this report is low; however, we think that our hypothesis of biosynthetic pathways and experiments leading to the determination of absolute configuration involving chemical transformations are of sufficient importance. Therefore, we believe that our report is in line with Marine Drugs Special Issue “Chemical Modification and Structural Elucidation of Marine Natural Products”.

For the structure-activity relationship (SAR)

As be pointed out, the amount of information is small to reach a sufficiently convincing conclusion.

We removed the description of SAR (page 7, line 175-179).

Round 2

Reviewer 3 Report

  1. This manuscript can be acceptable by Marine Drugs after necessary editing of English language. For instance, lines 15-16, the sentence "As the unique structural feature, heterocycle as diketopiperazine derivative and sugar conjugated to the 14-membered rings of them" is inappropriated in grammar.
  2. The title needs a further revision. As the hypothesis of biosynthetic pathway of halosmysins A and B is too limited, it should not be emphasized in the title. Thus, two options "Isolation and structure elucidation of new cytotoxic macrolides halosmysins B and C from the fungus Halosphaeriaceae sp. associated with a marine alga" and "Isolation and structure elucidation of new cytotoxic macrolides halosmysins B and C" would be suggested for consideration.
  3. Line 293, "very potent" should be revised as "active" because it can not be called "potent" with IC50 values of 8.2 and 20.5 micromolar in the in vitro cytotoxicity assay against selected tumor cell lines.

Author Response

Comments of Reviewer 3

1. This manuscript can be acceptable by Marine Drugs after necessary editing of English language. For instance, lines 15-16, the sentence "As the unique structural feature, heterocycle as diketopiperazine derivative and sugar conjugated to the 14-membered rings of them" is inappropriated in grammar.

2. The title needs a further revision. As the hypothesis of biosynthetic pathway of halosmysins A and B is too limited, it should not be emphasized in the title. Thus, two options "Isolation and structure elucidation of new cytotoxic macrolides halosmysins B and C from the fungus Halosphaeriaceae sp. associated with a marine alga" and "Isolation and structure elucidation of new cytotoxic macrolides halosmysins B and C" would be suggested for consideration.

3. Line 293, "very potent" should be revised as "active" because it can not be called "potent" with IC50 values of 8.2 and 20.5 micromolar in the in vitro cytotoxicity assay against selected tumor cell lines.

Reply to the review report

  1. Thank you for your pointed out. We corrected the sentence (line 15-16) as below.

“As the unique structural feature, heterocycle as diketopiperazine derivative or sugar are conjugated to the 14-membered rings of them.”

  1. We changed the title according to the suggestion of other reviewer, but as you pointed out, we think he hypothesis of biosynthetic pathway is limited. Therefore, we adopted the former of reviewer’s proposal, and changed the title as follows.

“Isolation and structure elucidation of new cytotoxic macrolides halosmysins B and C from the fungus Halosphaeriaceae sp. associated with a marine alga”

  1. As be pointed out, In line 293, we changed " very potent" to “active”.